# Perspective on the Prevention of Suicide among School Learners by School Management

**DOI:** 10.3390/ijerph20105856

**Published:** 2023-05-18

**Authors:** Hilda N. Shilubane, Robert A. C. Ruiter, Lunic B. Khoza, Bart H. W. van den Borne

**Affiliations:** 1Department of Advanced Nursing Science, University of Venda, Private Bag X, Thohoyandou 0950, South Africa; 2Department of Work and Social Psychology, Maastricht University, Minderbroedersberg 6, 6211 LK Maastricht, The Netherlands; 3Department of Health Studies, University of South Africa, Pretoria 0003, South Africa; 4Department of Health Promotion, Maastricht University, Minderbroedersberg 6, 6211 LK Maastricht, The Netherlands

**Keywords:** role, school managers, teachers, school, suicide

## Abstract

Background: Adolescents in South Africa have higher suicide rates than older people. A suicide or unexpected death of a fellow student can result in increased copycat behavior. Previous studies have placed emphasis on the significance of school involvement in the prevention of suicide. The study sought to explore the perspective on the prevention of suicide among school learners by school management. A qualitative phenomenological design was applied. The study used purposive sampling to select six high schools. In-depth interviews were conducted with six focus group discussions comprising fifty school management. A semi-structured interview guide guided the interviews. Data were analyzed using a general inductive approach. Findings revealed that school management should be supported through workshops to increase their skills in handling stressful situations at school. Support for learners through audio-visuals, professional counseling, and awareness campaigns also emerged. Parents–school partnership was said to be effective in preventing suicide among learners as both parties will be free to discuss the problems faced by the learner. In conclusion, empowering school management in the prevention of suicide is critical for Limpopo learners. Awareness campaigns conducted by suicide survivors where they can share their testimonies is necessary. School-based professional counseling services should be established to benefit all learners, particularly those experiencing financial challenges. Pamphlets in local languages should be developed for students to convey information about suicide.

## 1. Introduction

Suicide among the youth is a key public health concern [1]. In South Africa, adolescents have higher suicide rates than adults and the aged [2]. A suicide or unexpected death of a student can result in increased contagion, meaning that suicide may trigger other students to commit suicide as well.

According to Marsh [3], teachers and school management should be empowered to be able to deal with the at-risk group, as well as peers after the death of a fellow learner by suicide. However, little is known about their perceptions regarding learner suicide prevention [4]. Studies demonstrate that teachers are often not involved in youth suicide matters, but they spend lots of time with learners and, therefore, can perform a vital part in the prevention of suicide in various ways before, during, and after death by suicide. Furthermore, they can implement procedures to prevent copycat behavior after the death of a learner by suicide, enhance students’ emotional well-being, reach out to the at-risk student, and refer them to appropriate mental health experts for assistance [5,6]. 

Preventing learner suicide is a concern even though prevention programs exist internationally, and most are in schools [7,8,9]. Among the available programs, only a few have demonstrated a reduction in suicide attempts and improved suicide prevention, for example, Question, Persuade, Refer (OPR) [10]. 

Schools are progressively being acknowledged as vital settings for addressing suicidality and engaging in the prevention of teenage self-destructive behaviors because pupils spend a significant amount of time in educational institutions [11,12]. Self-destructive inhibition and mediation initiatives in schools are vital methods in attempting to address youth suicide and encouraging students to seek help [12]. Identifying at-risk students early is critical to providing effective suicide interventions, and school administrators contribute significantly to the reduction of learner suicide because they spend the majority of their time with students.

Irrespective of their recognition as important contexts in suicide prevention, schools are often uncertain of their position in reducing the risk of suicide and how to react when contacted by a pupil [13]. Therefore, well-informed school management can act as a source of support for grieving students who lost a fellow student to suicide and find it challenging to deal with the loss [14]. Although school management should help learners cope with death, unfortunately, they often feel unprepared [15].

Schools can be a good place to offer essential services to pupils after a loss that affects the entire school since large numbers of students can be served and their ability to cope after the loss can be monitored through referrals [15]. It is important to talk about the connection between depression and suicide after the death of a learner by suicide. This may help friends and loved ones to understand why it happened. Bereavement is a common experience among learners, but unfortunately, the role of school management and staff is neglected in this area, and they often receive little or no training [16].

Suicide prevention programs in South Africa are almost non-existent and are limited to two programs that were developed and implemented by the South African Depression and Anxiety Group (SADAG) in some parts of the country. The first program is an educational workshop on depression and suicide, and the second is an audiobook that aimed to bring suicide out of the taboo context. The programs were found to improve students’ understanding of depression and suicide. However, the challenge with these programs was that they were not implemented in all the provinces and the effects were seen in a very short time following the intervention [17]. Furthermore, there is a lack and non-use of such programs in Limpopo province where the current study took place. The current study sought to explore the perspective on the prevention of suicide among school learners by the school management (people responsible for the day-to-day activities that happen in the school).

## 2. Methods

### 2.1. Research Design

A qualitative, phenomenological design was used to explore school management’s perspectives on the prevention of suicide among school learners. Qualitative research seeks to understand social phenomena through exploration and interpretation of the meaning participants attach to a phenomenon [18]. This design assisted the researchers in building a complete picture of participants’ perspectives on the prevention of suicide among school learners. 

### 2.2. Participants and Sampling

The population was school management who voluntarily took part in the research. Six schools were purposefully sampled in Limpopo province, and participants were purposefully selected because they were managers and had teaching experience. These managers did not have training in mental health or suicide prevention. Furthermore, this was their first time participating in this study.

Three of the six schools lost a learner to suicide. It is worth explaining that during the process of sampling, the first author did not know whether the school lost a learner to suicide or not. This was revealed in the course of group discussions. The first author used the opportunity when visiting schools to meet the principals for permission to invite participants. Upon arrangement and with approval given to conduct the study, the principal of each school requested school management to partake in the focused discussion. The first author did not indicate the number of participants to be included in each interview since enrollment of learners dictates the size of the management team and she did not want to leave others out. The first author was given the number of management teams in each school, which ranged from six to thirteen based on the enrollment. All participants (50) given to the first author participated in the study. The aim of the study was clarified, and approval was obtained before participants participated in the discussion groups. 

### 2.3. Research Instrument

An information-gathering interview guide was used to explore the perspective of school management concerning suicide prevention among school learners. The interview questions that guided the discussions were generated from the literature consulted and a qualitative study on high school suicide in South Africa [19]. The main questions were, what kind of support does school management need to prevent suicide among school learners? What kind of support do school learners need to prevent them from committing suicide? The instrument was pre-tested with two focus group discussions with school management comparable to the participants in the study from other schools to ensure clarity and suitability of questions to participants. The two groups did not form part of the study. All questions were clearly understood, and the instrument was therefore not revised.

### 2.4. Data Collection Procedures

Focus group discussions with fifty school management personnel were conducted, of which twenty-seven (27) were females and twenty-three (23) were males. The interviews were conducted at schools during school breaks and ranged from 35–60 min. The interviews aided by an interview guide were conducted until data saturation was reached as demonstrated by the redundancy of the themes. Participants gave informed consent prior to conducting interviews, and participants were informed about voluntary participation and withdrawal at any time without penalty. The English language was used by the first author when conducting interviews. The interviews, which were voice-recorded with participants’ permission, were transcribed word for word. The first author facilitated the discussion groups that comprised six to thirteen participants each. The first author was the only one to conduct the interviews, ensuring confidentiality and privacy for all participants. Verbatim quotations are used for readers to judge the appropriateness of the study. It should be noted that the interviewer had no relationship with the participants prior to study commencement and no repeat interviews were conducted.

### 2.5. Trustworthiness

The face-to-face interaction with the participants promoted trustworthiness. Long-term engagement with participants while filling out consent forms and conducting interviews ensured credibility. The thorough description of the research method ensured transferability. The researcher ensured dependability by use of field notes and a voice recorder to guarantee the correctness and by availing the transcripts for authentication if needed; and by peer examination by co-authors who assessed the correctness of the methodology undertaken. The discussions were conducted until a point where no new information was discovered in the analysis and an audit trail was developed to ensure confirmability. The documentation consisted of field notes and transcripts [20].

### 2.6. Data Analysis

The audio-recorded data were transcribed before data analysis was undertaken. The first author read and checked the interviews for accuracy. The thematic analysis method was used to analyze data collected through open-ended research questions. An inductive approach, which involves the search for patterns from observation and the development of explanations, was applied. The researcher recognized topics arising from the data, rather than trying to fit them into a prior coding frame. Initial coding was undertaken independently by the first and second authors. The two authors reached an agreement on the coded items. The coded items were then assembled into larger main themes and sub-themes. Member check was performed with participants. The authors conducted literature control by comparing the current study findings with existing research, which permitted the results of the study to be contextualized within universal scientific knowledge without any due influence of that knowledge [21].

### 2.7. Ethical Considerations

The Ethics Committee of the University gave approval (Project Number: SHS/16/PDC/15). The Provincial Department of Education together with the head of the concerned schools in Mopani, Vhembe, and Capricorn Districts granted permission for the study to be carried out. Participants gave informed permission before the interviews and were informed that involvement in the study was of one’s own free will and that confidentiality would be maintained. They were informed that they could withdraw if they so wished without explanation and that there were no consequences if they chose to withdraw. Participants were guaranteed that their answers would not be attached to their identities since no names would be used. They were informed that records would be kept for three years before being destroyed to adhere to the principles of record keeping.

## 3. Results

### 3.1. Demographic Characteristics

Fifty participants (27 females and 23 males) participated in the study. The majority had six and more years of experience in teaching and managing the school. Of the 50 participants, only 2 changed their careers; the first one was a qualified educator but could not obtain a job and had to work as a cleaner before being appointed as an educator. The second participant was working at the independent election commission but decided to enroll for a degree in teaching that opened the way for his career.

### 3.2. Themes and Sub-Themes

The data revealed two themes and two subthemes, which are presented below in Table 1. 

#### 3.2.1. Support Strategies

The theme emerged with two sub-themes, support for school management and support for learners.

##### Sub-Theme 1: Support for School Management

Participants mentioned during the discussions that they need experts in the field to conduct workshops. They thought empowerment would make them confident and comfortable to discuss suicide with learners and be able to handle both the school employees and learners in case of a death by suicide. Participants further mentioned that establishing school-primary health care forums with the local clinics for educating staff on the suicidal symptomatic behavior of learners would be beneficial. 

The following quotes attest to this.

I think if teachers can regularly have workshops, they can have the zeal to face learners who lost a peer through suicide as well as deal with fellow teachers.

Workshops will be most useful. As we have never experienced a situation where a learner passed on as a result of suicide, we won’t know how to handle such a situation until someone with experience can workshop us.

Another suggestion is to introduce a strong school-clinics relationship so that health workers could freely do school visit screens and counsel learners who show suspicious behavior. It should be taken into consideration that at least one clinic staff be part of the school governing body as an ex-officio member.

Another participant had a different view and thought a suicide survivor is the best person to workshop them.

I think having a person who attempted suicide tell his or her story can bear more fruits. Some of us believe in someone who experienced the situation more than any other person. This person will share even the problems that led to the act, unlike the one who gets the information from books.

##### Sub-Theme 1.2: Support for Learners

Participants indicated that educating learners could also benefit the community as learners could, in turn, educate their family members and members of the community on suicide-related behavior and also identify those with threatening symptoms of suicide. In addition, they perceived the following as important to enhance the support for learners. 


*Reading Material*


It was revealed during the interviews that schools do receive various documents from the government to educate learners beyond the classroom; however, educational documents related to mental health, signs and symptoms of suicidal behavior, and suicide prevention were never received. Participants mentioned that pamphlets written in the local language could assist the learners as most learners do not understand English. Participants gave an example of HIV/AIDS pamphlets that have been delivered to their schools and had not been given to learners because they do not have time to read and explain to them since learners will not understand those pamphlets without the teachers’ explanations. The participants in different focus groups had this to say,

If we could have posters in class could serve as awareness to our learners. Each time they see the poster it will remind them that this is something that can affect them.

Some mentioned the issue of using a language understood by young people and indicated that in writing leaflets with suicide-related information, a local language is preferred.

Pamphlets on causes of suicide and manuals on problems faced by teenagers can assist. These should be written in the mother tongue of the learners, for example, the mother tongue of our learners is Sesotho, which means the manuals should be written in Sesotho.

The materials should be in their home language because when written in English they will perceive them as if are the things of the media like the newspaper. If written in their mother tongue it will arouse their interest to look at the document and also read it.

In addition, some participants came up with the idea of including pictures of hopeless individuals on the leaflets with suicide information.

I think to curb this problem, booklets that contain detailed information on suicide should be available for learners. Those booklets should have pictures showing someone suicidal.


*Audio-Visual Material*


Participants mentioned that having audio-visuals can assist in preventing learner suicides. The videos could be played to arouse their interest before the teacher could deliver a lecture. The following quote attests to this.

Audiovisuals can also assist because during life orientation lessons one can first play a video that talks of suicide-related behaviors before presenting a lecture to them.

In another group, a participant said:

Although we are rural, some learners are gifted and can perform role play in the absence of videos, and in that way, it can benefit those learners who learn by seeing. 

Another participant said:

Soul buddies’ books are required at school, when they see them on TV, they think is something that happens in Gauteng only, but if they can read it, they will realize that it can also affect them. Also, doing drama at school focusing on a specific topic can assist.


*Professional Counselling*


Participants revealed that counseling is required, and the service should be accessible to both school management and the entire school community when needed. These include the services of social workers, psychologists, and pastors.

Professional Counselors like psychologists and pastors are needed. The government should bring the service of social workers to us as some learners don’t have money to go and see a social worker. For example, you might find that a fellow learner knew that a friend had problems but failed to share with someone, after the suicide death the fellow learner could be affected and may need counselling from social workers.

We need a social worker to be available to counsel them. Again, we need support for learners since most of the parents are working far away from home leaving the learners heading the families. We really need support, but I don’t know how government can assist us.

The services of the pastor were perceived to be crucial. 

Since there is no religious education in schools, the school can invite pastors to come and have sessions with learners, sessions like all ladies where teachers teach the young girls about puberty and what to do when it arrives.


*Awareness Campaigns*


Awareness campaigns can be effective in providing information to learners and the community. Participants thought it would be great to involve suicide survivors (family members who lost a loved one by suicide) during the campaign so they can share their personal stories to increase public awareness and education and to reduce further suicides. 

I perceive roadshows and learner competitions as effective tools in preventing the suicide of learners. Suicide survivors are the relevant people for roadshows.

Some participants in the group discussions saw nurses’ role as a solution for this.

I grew up seeing nurses taking the lead in educating people, particularly learners. During our time they were going to schools to educate learners on health-related issues. Where are the nurses today because we need them? 

#### 3.2.2. Family–School Partnership

The family is the primary support system for the learner, and participants saw the involvement of parents in the education of their children as crucial. Participants expressed a need for a good relationship between parents/guardians and schoolteachers. Such a relationship could promote openness where parents could freely communicate their problems to school staff who could take an appropriate intervention. 

There is a need for communication between teachers and parents/guardians. At times the problems that learners have are from home and as school management, we are not aware of such problems. Good communication between the two parties could help wherein the parents could inform management of such problems.

What we need is a good relationship between us and the parents for us to identify what could be the cause of this suicide. You may find that there are problems in the family which in turn affects the learner, and the learner does not want to share the problems with anyone which may lead to suicide. If there is intimacy between us and the parents, we would know learners more, so we need support from the parents and the Department. We cannot turn a blind eye as if these things are not happening.

Some participants saw parents as a support for the learners. The excerpt below supports this statement,

Parents should do their role of teaching and disciplining their children since schools are not allowed to discipline learners anymore. I still blame the government because it brought the rights but did not bring the solution to them.

## 4. Discussion

Participants in three focus group discussions mentioned that they lost a learner through suicide, although there was a tendency to forget that it happened. When asked what information they gave to learners, they mentioned that they told the deceased’s classmates that it happened and they cannot change it; therefore, they should accept it and life must go on. They mentioned that they took someone to approach the learners who shared a class with a learner who passed on. Their hesitance to confront them could be that they did not know what kind of message should be communicated to the learners as demonstrated by the need for experts to workshop them.

Research demonstrates that in case of death, the responsible school committee should perform its function of addressing the school community, which was not the case in this study as they did not have such a team [22]. The school management revealed a lack of skill to handle a school crisis. This could be the reason that made participants want to be empowered through workshops. Therefore, workshops could be a strategy that could assist in the prevention of suicidal behavior among school learners through educators’ ability to identify warning signs of suicidal behavior. Since school management spends many hours with the students, empowering them could decrease the scourge as they would empower their subordinates, thus responding appropriately to at-risk students [23,24]. Workshops can also play an important role after the death of a learner by suicide.

Although school management mentioned that they addressed the learners and visited the deceased’s home to offer support through prayers, it did not seem to help learners as they continued to demonstrate fear and could not concentrate during teaching and learning, which led to poor teaching and learning in class. The pastor’s counseling service seemed to have comforted the learners. This is in line with studies that showed religion plays an important role in the protection and coping after suicide [25,26,27].

It is not surprising to hear that those learners were uncontrollable, an unexpected death causes frustrations and psychological pain. When a suicide occurs, youth can be deeply affected and become vulnerable to suicide contagion if not handled properly. It is essential to support students’ coping postvention. This entails giving support to students after a suicide by a peer [3,28]. The degree of impact differs depending on the age of the student, the time of the year, whether the student died on- or off-campus, and the number of family members who attended the school of the deceased. Thus, providing immediate and long-term support to grieving learners is essential.

Enhancing learners’ coping skills through pamphlets and audio-visuals could benefit them. A study conducted by Wasserman [29] in “European Union Countries found that the Youth Aware of Mental Health Programme (YAM) which raises pupils’ awareness and ability to cope with depression, anxiety and suicidality” was efficient in lowering the risk of suicide when compared with “Question, Persuade, and Refer (QPR) and gatekeeper training program for school personnel”. Providing learners with pamphlets, journals, and videos in their school libraries could effectively prevent or reduce the suicide rate in schools and the community by enhancing their problem-solving skills and making them aware of available services of professionals at schools or nearby.

Participants raised a concern about child-headed families due to parents working far away or the death of both parents; therefore, the availability of social workers at schools could reduce the suicide rate because at times learners need such services but cannot access them because of the distance and cannot afford to board a bus or taxi to reach them due to financial constraints.

According to one study [30], personal testimonies are inspirational as well as educational for the public and others bereaved by suicide. Considering that suicide is something not to be discussed openly in some societies, outreach efforts by survivors of suicide to raise awareness of suicide could be beneficial. Furthermore, a genuinely open discussion, not just medical and health discussions, is crucial [31]. Even though such discussions are necessary, caution should be taken to avoid a contagion effect; organizers should provide support to suicide survivors by offering to act as an audience for them to rehearse.

For suicide prevention among learners to be a success, school management must incorporate stakeholders [32]. Family involvement in schools benefits both students and schools by increasing student achievement and attendance while also fostering students’ emotional and social development. Furthermore, when schools communicate with parents about school activities and programs, students feel more competent, and both students and parents are more likely to work to keep those activities and programs going [33]. Furthermore, students and staff can support academic progress and enhance their total well-being [34].

After school, parents and guardians are the ones who spend the remainder of the day with learners. Their involvement is critical and can assist in reducing the self-harm of learners and promoting good relationships with the staff. Family partnerships are important in moral regeneration and nurturing both the emotional and academic success of the learners [26]. Empowering parents on suicide could assist in identifying and responding to signs of poor mental health as well as creating a safe environment for at-risk learners. Furthermore, it can promote a good relationship between parents and schools.

## 5. Conclusions and Implications for Practice

South Africa has a scarcity of data on the perspective prevention of suicide among school learners by the school management. Nonetheless, the study’s goal was to provide insights into the school management’s viewpoints regarding the prevention of learners’ suicide. Even though the current study has limited generalizability due to the specific geographic areas studied, its subjective nature, and the years of experience in their current employment as the only sociodemographic characteristic of the participants collected, the current study findings suggest that empowering school management in the prevention of suicide is critical for Limpopo learners. The study demonstrates that learners cannot read documents presented to them because of the language used. Pamphlets in local languages should be developed for students to convey information about suicide. The outreach campaigns by suicide survivors and open discussions of suicide, not just medical or health discussions, are essential. As most suicide campaigns focus on medical and health discussions, open discussions that include social, cultural, emotional, and political aspects of suicide could be an effective strategy for suicide prevention among learners. Mental health nurses, social workers, and psychologists are examples of professionals who could reinforce school lessons on mental health promotion. This act might promote collaboration between departments and get rid of the fragmentation of services. School-based professional counseling services should be established to benefit all learners, particularly those experiencing financial challenges. The study’s findings are a precursor to future research, particularly quantitative research to ensure the representative and generalizability of the findings.

## Figures and Tables

**Table 1 ijerph-20-05856-t001:** Themes, subthemes, and categories.

Themes	Sub-Themes	Categories
3.2.1 Support Strategies	3.2.1.1 Support for school management	
	3.2.1.2 Support for learners	3.2.1.2.1 Reading material 3.2.1.2.2 Audio-visual material 3.2.1.2.3 Professional counselling 3.2.1.2.4 Awareness campaign
3.2.2. Family-School Partnership		

## Data Availability

The data presented in this study are available from the corresponding author upon request. The data are not publicly available because they contain the names of schools that participated in the study.

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
