# Peer review of "Perspective on the Prevention of Suicide among School Learners by School Management"

_ijerph, 2023, doi:10.3390/ijerph20105856_

Round 1

Reviewer 1 Report

The current manuscript relates to a qualitative study which explores the perception of the school managers. I believe that this type of work is commendable because it could be helpful to explore implementation and adoption barriers, however I believe that the current of version of the manuscript requires substantial work.

Particularly, the whole manuscript need to specify the theoretical framework which will support the interpretation of the data, or if the authors already have one, it requires to specify it.

In the introduction, the description of the context of the problems, is quite right. However it requires to describe what type of information the will look for in their study. For instance, there is a framework called implementation science that focuses on the barriers and facilitators of the implementation and adoption processes of the prevention programs. However this manuscript is unclear regarding their specific theoretical approach.

This problem also applies to the description of the instrument. Please state which were the questions or subjects that were planned in the interview. Also, describe with more detail the literature that was considered for the instrument. Also, please describe which method was used to pilot the interview.

Please consider including any relevant information when describing the sample. Information such as the previous training in mental health or suicide prevention, and previous experiences regarding the subject could be useful.

Please describe with more detail how many researchers participate in the coding and interpretation process. Also describe if there were any kind of triangulation between coders.

Please describe how many participants were recruited and how many were analyzed. If there was loss in any part of the study, consider including a flow chart.

Table 1 is a little messy, I suggest to better use a diagram o display It in a paragraph.

Please consider that it is hard to stablish a critical appraise about the conclusions and discussion as the objectives seems to broad to compare the information.

How the data of this study compares to previous studies?

What were the main limitations of the study?

Author Response

Dear reviewer,

Kindly find our responses for your attention.

Kind regards,

Hilda

Reviewer 2 Report

First of all, thank you for giving the opportunity to review this interesting paper, with a really relevance topic.

In genera, it is well explained and structured.

However, I find some points that could be improved:

Abstract

In line 24, this sentence “In conclusion, since the study demonstrates that learners cannot read documents pre-24 sented to them on suicidal behaviours because were written in English language, local languages 25 should be used to develop suicide pamphlets.” I don't think it is correct, since it is not the conclusion of the study, or at least the only conclusion.

Introduction

The introduction is well documented and structured. It adequately explains the current problem of suicide among school learners.

Methods

Did you collect more sociodemographic characteristics of the participants?

Themes and sub-themes subsection is not clear.

First of all, the table has no title and the structure of the table is not well understood.

It seems that there are two sections:

1. Support strategies

2. Family-School Partnership

Within Support strategies there are two other subsections:

1.1 Support for school management

1.2 Support for learners

This information should be more clearly reflected both in the table and in the text (such as proper subsection numbering).

Regarding the content, it seems well explained

Discussion

An adequate discussion of the results is carried out

Author Response

Dear Reviewer,

Reviewer 3 Report

The topic of prevention has received relatively little attention in the scholarly literature on suicide. The authors of the article attempt to fill this gap by offering an intriguing viewpoint on the issue, which has increased in recent years. They demonstrate how complicated and troubling the subject of suicide is, particularly in the context of schoolchildren and teenagers, and they also provide a novel solution that focuses specifically on the assistance aspect of the problem.

The text's shortcomings, in my opinion, are a slightly disjointed and unclear argument as well as a lack of a clear rationale for the selection of the research method.

The text demonstrates a thorough understanding of the relevant literature, and the scientific tools are correctly applied. After the necessary corrections, the article will satisfy the formal specifications for publications related to the chosen topic, but I advise that the last section of the text should clearly state that the study's findings are a precursor to future research work, based on a broader methodological workshop (for example, the method of triangulation), which I believe is necessary.

Author Response

Dear Reviewer,

Reviewer 4 Report

Questions and comments for consideration:

1.     What exactly is a school manager? Is such an individual an administrator or would another term be more enlightening?

2.     What is inductive data analysis? Does it differ from inductive reasoning?

3.     Are there hypotheses regarding how opinions of school managers from volunteer schools might differ from the opinions of individuals from non-volunteer schools?

4.     Is there an advantage to labeling high school children/adolescents as school learners rather than simply “students?”

This study seems to adhere to standards of phenomenological research: It is descriptive of the phenomenon of perspectives on suicide prevention among high school students in South Africa, at least from the perspectives of a subset of “school managers.”  Further, as required in the methodology, the authors attempted adequately to integrate their interview-related themes with extant theory and research in suicide prevention.

My critique relates to (a) the “ho-hum,” prosaic nature of the findings and (b) the deficient external validity of the study; the extent to which the findings may generalize beyond a province in South Africa

First, concerning point “a,” the conclusion of the study (abstract, lines 24 to 27) is that documents about suicide should be printed in a language that can be read by students and school personnel.  This is hardly more than an elucidation of the obvious and is not worthy of publication. Additional conclusions mentioned in the body of the article include increasing availability of relevant information from mental health and medical professionals, testimonials of suicide survivors, providing access to the best available information through workshops, written documents, and audio visual aids together with teacher and student support from the most qualified people is nothing beyond good common sense, available to any thoughtful exclusive of this phenomenological study (see lines 355-367). 

Second, concerning point “b,” the participants were 50 “school managers” selected from six “volunteer” high schools in one South African province. These particular high schools were selected presumably (because it is not explicated in the article) from a larger pool of volunteer high schools. The first external validity question is how the opinions of school volunteers might differ from the opinions of non-volunteers in answers to the phenomenological questions. Perhaps the latter would opine concepts that deviated from common sense. Regardless, the degree of generalization from the voluntary to the involuntary is simply unknown. The other unaddressed issue related to external validity involves terminology that may be unique to the South African educational system or to the authors and may lack meaning in other geographic areas.: What is a “school manager” for example? What is “inductive data analysis?”

Author Response

Dear Reviewer,

Please see the attachment for your attention.

Reviewer 5 Report

This article needs to be improved in several ways.

First of all, the literature on public health and psychiatry which is relevant to understanding issues concerning suicide (ideation/self-harm/potentially lethal self-harm/ completed suicide) that is relevant for South Africa needs to be reviewed in detail, in order to help the reader understand the relevant cultural, social and clinical background factors which "school counsellors" (?managers) should be aware of in preparing themselves as suicide prevention agents in their schools.

The literature review offered by this article is sparse, and I have appended a number of references which should be consulted and summarised in the initial section (the authors cite, but do not expand on their own work).

The qualitative, focus group strategies are adequate, but do indicate a rather dismal' lack of professional knowledge and commitment. Using "managers" as potential suicide prevention agents is problematic. Surely managers are engaged in other tasks than counselling. It should be frontline workers (teachers and counsellors), surely who should have been selected for study? This needs clarifying.

The authors should have made some attempt to establish figures for completed suicide of pupils in the previous five years in the schools studied. Surely these supposedly catastrophic events are not forgotten entirely?

It would be useful to know the number of pupils in each class in the schools studied (many pupils would make it difficult for a teacher to have knowledge of individual pupils)? What is the language of instruction used by teachers - English, or the majority local language of Sesotho? This is relevant in considering how teachers can act as suicide prevention counsellors.

It's puzzling that the important article by Kootbodien et al (2020) published in the journal to which the paper under review  is being submitted to, has been ignored. Limpopo has a low recorded rate of suicide at 6.7 per 100K, compared with that of 31.5 per 100K in Natal Province. This could mean that completed suicide rate in Limpopo students is rare: this needs to be clarified. For the article under review to have importance, it should offer a commentary on new and relevant programmes for  all of South Africa. Is it true, as the press article I have copied, that continued poverty is a major cause of suicide?

https://www.enca.com/life/59-teenage-suicides-limpopo-2021

https://mg.co.za/opinion/2021-10-05-suicide-crisis-soars-in-south-africa/

Kootbodien, T., Naicker, N., Wilson, K. S., Ramesar, R., & London, L. (2020). Trends in suicide mortality in South Africa, 1997 to 2016. International journal of environmental research and public health17(6), 1850.

Alabi, A. A. (2022). Self-confidence and knowledge of suicide assessment and prevention amongst first-line health professionals in Nelson Mandela Bay, South Africa. South African Family Practice64(1).

Shilubane, H. N., Bos, A. E., Ruiter, R. A., van den Borne, B., & Reddy, P. S. (2015). High school suicide in South Africa: teachers’ knowledge, views and training needs. BMC public health15, 1-8. [This is by the authors of the paper under review: these earlier findings should be expounded in detail, as establishing the basis for the qualitative design]

Bantjes, J., Swartz, L., & Cembi, S. (2018). “Our lifestyle is a mix-match”: Traditional healers talk about suicide and suicide prevention in South Africa. Transcultural psychiatry55(1), 73-93.

Mngoma, N. F., Ayonrinde, O. A., Fergus, S., Jeeves, A. H., & Jolly, R. J. (2021). Distress, desperation and despair: anxiety, depression and suicidality among rural South African youth. International review of psychiatry33(1-2), 64-74.

Khuzwayo, N., Taylor, M., & Connolly, C. (2018). High risk of suicide among high-school learners in uMgungundlovu District, KwaZulu-Natal Province, South Africa. South African Medical Journal108(6), 517-523.

Thornton, V. J., Asanbe, C. B., & Denton, E. G. D. (2019). Clinical risk factors among youth at high risk for suicide in South Africa and Guyana. Depression and anxiety36(5), 423-432.

Burrows, S., & Laflamme, L. (2008). Suicide among urban South African adolescents. International journal of adolescent medicine and health20(4), 519-528.

Mngoma, N. F., & Ayonrinde, O. A. (2022). Mental distress and substance use among rural Black South African youth who are not in employment, education or training (NEET). International Journal of Social Psychiatry, 00207640221114252.

De Wet, N. (2017). Gendered risk factors associated with self-harm mortality among youth in South Africa, 2006-2014. South African Medical Journal107(12), 1132-1136.

Cluver, L., Orkin, M., Boyes, M. E., & Sherr, L. (2015). Child and adolescent suicide attempts, suicidal behavior, and adverse childhood experiences in South Africa: a prospective study. Journal of Adolescent Health57(1), 52-59.

Buckley, J., Otwombe, K., Joyce, C., Leshabane, G., Hornschuh, S., Hlongwane, K., ... & Violari, A. (2020). Mental health of adolescents in the era of antiretroviral therapy: is there a difference between HIV-infected and uninfected youth in South Africa?. Journal of Adolescent Health67(1), 76-83.

Smith, Z. (2021). Death due to hanging: a retrospective descriptive study of the socioeconomic and demographic profiles of hanging victims in central South Africa. Forensic Science, Medicine and Pathology17(2), 223-229.

This is not an exhaustive list: the authors should supplement this with their own search of documents and journals.

Author Response

Dear Reviewer,

Round 2

Reviewer 4 Report

Your responses to reviewer criticisms and resultant modifications warrant publication of the manuscript.

Reviewer 5 Report

The paper has been improved by clarifying the role of "managers". The qualitative design does clearly identify ways in which these managers can address suicidal issues experienced by their students. Uncoupling the design from an epidemiological framework has resolved ambiguities about the study's relevance. It now stands alone as a qualitative study which can be replicated in other domains of training and education for school counsellors.